# The Regenerative Effect of Portal Vein Injection of Liver Organoids by Retrorsine/Partial Hepatectomy in Rats

**DOI:** 10.3390/ijms21010178

**Published:** 2019-12-26

**Authors:** Tomonori Tsuchida, Soichiro Murata, Koichiro Matsuki, Akihiro Mori, Megumi Matsuo, Satoshi Mikami, Satoshi Okamoto, Yasuharu Ueno, Tomomi Tadokoro, Yun-Wen Zheng, Hideki Taniguchi

**Affiliations:** 1Department of Regenerative Medicine, Yokohama City University Graduate School of Medicine, 3-9 Fukuura, Kanazawa-ku, Yokohama 236-0004, Japan; t126040f@yokohama-cu.ac.jp (T.T.); t176521c@yokohama-cu.ac.jp (K.M.); t176065b@yokohama-cu.ac.jp (A.M.); mmatsuo@yokohama-cu.ac.jp (M.M.); smik@yokohama-cu.ac.jp (S.M.); sokamoto@yokohama-cu.ac.jp (S.O.); uenoya@ims.u-tokyo.ac.jp (Y.U.); tadokoro@yokohama-cu.ac.jp (T.T.); ywzheng@md.tsukuba.ac.jp (Y.-W.Z.); 2Division of Regenerative Medicine, Center for Stem Cell Biology and Regenerative Medicine, The Institute of Medical Science, the University of Tokyo, 4-6-1 Shirokanedai, Minato-ku, Tokyo 108-8639, Japan; 3Department of Surgery, Faculty of Medicine, University of Tsukuba, 1-1-1 Tennodai, Tsukuba, Ibaraki 305-8575, Japan

**Keywords:** stem cell therapy, organoid, xenotransplantation, fetal liver cell, human iPS cell

## Abstract

In this study, we reveal that liver organoid transplantation through the portal vein is a safe and effective method for the treatment of chronic liver damage. The liver organoids significantly reconstituted the hepatocytes; hence, the liver was significantly enlarged in this group, compared to the monolayer cell transplantation group in the retrorsine/partial hepatectomy (RS/PH) model. In the liver organoid transplantation group, the bile ducts were located in the donor area and connected to the recipient bile ducts. Thus, the rate of bile reconstruction in the liver was significantly higher compared to that in the monolayer group. By transplanting liver organoids, we saw a level of 70% replacement of the damaged liver. Consequently, in the transplantation group, diminished ductular reaction and a decrease of placental glutathione S-transferase (GST-p) precancerous lesions were observed. After trans-portal injection, the human induced pluripotent stem cell (hiPSC)-derived liver organoids revealed no translocation outside the liver; in contrast, the monolayer cells had spread to the lungs. The hiPSC-derived liver organoids were attached to the liver in the immunodeficient RS/PH rats. This study clearly demonstrates that liver organoid transplantation through the portal vein is a safe and effective method for the treatment of chronic liver damage in rats.

## 1. Introduction

The liver has the ability to replace tissue lost after surgical resection, necrosis, or other injuries, and the restoration of damaged or resected liver tissue occurs through the proliferation of mature hepatocytes located in the remaining viable tissue [1]. Currently, the restoration of mass and function in severely damaged livers is based on the transplantation of hepatocytes. Hepatocyte transplantation has been evaluated in numerous clinical trials [2,3,4]; however, to date, its long-term efficacy remains unclear. Additionally, the paucity of donor cells represents a limitation to this strategy. Improvements in the long-term efficacy of cell transplantation can be achieved by using stem or progenitor cells derived from embryonic stem cells, induced pluripotent stem cells (iPSCs), and fetal liver cells. In a rat experiment, Dabeva et al. identified embryonal day 14 (ED14) fetal liver cells as epithelial cells, characterized by the ability to differentiate into adult liver tissue [5]. In another study, Oertel et al. named the ED14 fetal liver cells as stem or progenitor cells [6], because they can differentiate into hepatocytes and bile ducts [7]. The liver develops from a condensed tissue mass, called the “liver bud”, before blood perfusion. In earlier studies, our colleagues mimicked this early organogenic event by developing a culture method to generate transplantable liver organoids from human iPSCs (hiPSCs) cultured with mesenchymal cells, endothelial cells, and endodermal progenitors [8,9,10]. 

The transplantation of hepatocytes from the portal vein may lead to portal vein embolization and massive liver necrosis [11,12]. Coagulation parameters should be monitored with hepatocyte transplantation for the prevention of portal vein thrombus, because hepatocytes have been shown to display a tissue factor-dependent procoagulant effect in vitro [13]. The liver organoids are larger than single cellular hepatocytes, and the risk of portal vein embolization would be increased by portal vein administration. According to several previous reports, organoids are digested or partially dispersed in a single-cell suspension before orthotopic cell transplantation into the liver [14,15,16]. The achievement of a method of transplantation into the liver of functional organoids is very important. However, only one report on organoid transplantation from the portal vein have been published so far [17]. 

The aim of the present study was to evaluate the safety and efficacy of liver organoid transplantation from the portal vein for the treatment of chronic liver damage.

## 2. Results

### 2.1. Liver Organoids Made from ED14 Fetal Liver Cells

The ED14 fetal livers of the F344 dipeptidyl peptidase IV (DPPIV)-positive rats were harvested, and fetal liver cells were cultivated in a round-bottom type culture plate (Elplasia). The cells were condensed into a single cluster within 4 h; within 24 h, they formed a single uniform liver organoid (approximately 70 μm) in each well (Figure 1A,B). Albumin, HNF4alpha, and CD31 were expressed at higher levels in the liver organoid culture compared to the monolayer culture (Figure 1C,D). Likewise, RT-PCR revealed higher levels of albumin, cytokeratin 19 (CK19), hepatocyte nuclear factor 4 (HNF4) alpha, and glucose 6 phosphatase (g6p) in the liver organoid culture (Figure 1D). 

### 2.2. Liver Organoid Transplantation Accelerates Liver Regeneration in the Retrorsine/Partial Hepatectomy Model

After retrorsine (RS) administration, a two-thirds partial hepatectomy was performed in the F344 DPPIV-negative rats; this was immediately followed by trans-portal transplantation of the liver organoids (3.0 × 10^3^ liver organoids, each liver organoid contains 5.0 × 10^3^ fetal liver cells) and the fetal liver cells (fetal liver cell equivalent of 3.0 × 10^3^ liver organoids). Interestingly, 30 days post-transplantation, no massive necrosis or portal vein embolization was observed in the liver organoids or the monolayer fetal liver cells. The repopulation rates post-transplantation were clearly observed via CD26/DPPIV staining. Thirty days after transplantation, the repopulation rate in the liver organoid group was higher than that in the monolayer cell group, statistical significance notwithstanding (Figure 2A,B). However, the liver weight/body weight ratio in the liver organoid group was significantly higher than those found in the monolayer and sham-operated groups (Figure 2C). These findings indicated that the retrorsine/partial hepatectomy (RS/PH) liver was repopulated with the liver organoid, and that liver regeneration was accelerated within 30 days of the RS/PH treatment. One of the reasons for the increased liver weight/body weight ratio is the enlargement of liver organoid-derived colony diameter. The diameter of the liver organoid-derived colony was larger than the monolayer cell-derived colony (Figure 2D). In the recipient area, the nuclei of the hepatocytes were enlarged, due to chronic damage from RS. However, the nuclei of the hepatocytes in the donor-derived areas were normal. The donor-derived liver tissue (CD26 positive) expressed HNF4 alpha and CK19 (Figure 2E). The bile ducts present in the donor area were connected to the recipient bile ducts (Figure 2E). A significantly higher ability for bile reconstruction was noted in the liver organoid group compared to the monolayer group (Figure 2F). The liver sinusoidal endothelial cells were observed both in the donor area and recipient area, which indicates that normal liver structure was reconstituted in the damaged liver around the donor area (Figure 2G). In the first 96 h post-transplantation, liver organoids disassembled and spread faster than monolayer culture around the sinusoidal area without portal thrombus (Figure 2H).

### 2.3. Liver Organoids Reconstructed Normal Liver Structure and Inhibit Precancerous Lesion Derived by Chronic Injury from Retrorsine/Partial Hepatectomy

Over 180 days after transplantation, we observed that the survival rate in the fetal liver cell-derived liver organoid transplantation group was significantly higher compared with the sham operation group (Figure 3A). The repopulation rate was increasing 30, 60, and 120 days post-transplantation, up to 70% (Figure 3B,C). In the sham group, ductular reaction was maintained as CK19-positive (CK19+) cholangioles (Figure 3D). On the contrary, in the transplantation group, the CK19-positive cholangioles almost diminished, and normal bile ducts were observed in the periportal area (Figure 3D). Following liver organoid transplantation, there is a significant decrease of the CK19+ area (Figure 3D,E). In the sham group, it is possible to observe GST-p lesions. The latter is known as precancerous lesions. On the contrary, we observed a significant decrease of these lesions in the transplantation group (Figure 3F,G).

### 2.4. Trans-Portal Injection of Human Induced Pluripotent Stem Cell Liver Organoids Do Not Translocate Outside Liver

In a previous study, we generated human iPS cell (hiPSC) liver organoids on multi-well plates (Elplasia) [18]. No deformity was observed after cultivating the hiPSC liver organoids, and the cells were condensed to a single liver organoid in the microwell (Figure 4A). After one day of culture, the mean diameter of the organoid was 130.1 μm (Figure 4B). To compare the bio-distribution, single hiPSC-derived cells and liver organoids were transplanted to adult NOD SCID mice through the portal vein. Five organs (liver, lung, brain, kidney, and spleen) were evaluated 5 minutes after the transplantation. The bio-distribution was evaluated via human Alu PCR and immunohistochemistry. The human liver organoid was found completely within the liver; in contrast, single hiPSC-derived cells were detected in the lungs (Figure 4C). A histological analysis of each organ revealed single human cells in the lung tissues of the single hiPSC-derived cells group, whereas no human cells were observed in the brain, kidney, or spleen in the human liver organoid group (Figure 4D). Thus, the human liver organoids remained in the liver without translocating to the other organs.

### 2.5. Human Induced Pluripotent Stem Cell Liver Organoids Attached and Reconstituted in the Immunodeficient Rat Liver

An hiPSC liver organoid can be maintained for 22 days. The shape of the hiPSC liver organoids were stable until 12 days (Figure 5A). Human albumin was detected in the supernatant of hiPSC liver organoid until 22 days (Figure 5B). Fourteen days post-transplantation, hepatic structures were observed in the liver parenchyma of the IL2rg KO rats treated with RS and 30% partial hepatectomy (Figure 5C). The CK8/18 and human albumin-positive hepatocyte clusters were observed in the hiPSC liver organoid transplantation group. Human CK19-positive bile ducts were observed near the hepatocyte cluster (Figure 5C). The 8 day culture hiPSC liver organoids demonstrated the highest concentration of human albumin 14 days post-transplantation (Figure 5D), which indicated that the 8 day culture hiPSC liver organoids have better attachment than 12 day culture organoids.

## 3. Discussion

In the present study, we showed the engraftment of rat liver organoids derived from fetal liver in an RS/ PH model of the rat. Liver organoid transplantation through the portal vein accelerates liver regeneration of a chronically damaged liver. Liver organoid transplantation also decreased the ductular reaction and precancerous lesions of the RS/PH liver. Liver organoid transplantation through the portal vein was found to be a safe technique with no translocation to the other organs, whereas single-cell injections from the portal vein resulted in the translocation of the cells to the lungs. The transplantation of hiPSC liver organoids from the portal vein led to their attachment and repopulation in the IL2rg KO rats, followed by the secretion of human albumin.

There are several reports on the effectiveness of the organoid culture [14,15,16,17]. According to several previous reports, organoids are digested or partially dispersed into single-cell suspensions before orthotopic cell transplantation into the liver [14,15,16]. The main reason for not injecting organoids directly in the portal vein is the fear of the development of a portal vein thrombus [11,12]. In the present study, we achieved the safe transplantation of a liver organoid through the portal vein without the occurrence of massive necrosis or portal vein thrombus. The transplanted liver organoids were around 70 μm in size, and the attached organoids occupied larger reconstructed areas in the RS/PH liver, whereas the monolayer cells occupied small areas in each attached cell. As a result, the liver weight/body weight ratio was recovered in the liver organoid group, but was low in the monolayer cell group. Bile reconstruction was increased in the liver organoid group, owing to the significantly higher expression level of *Ck19* in this group when compared with the monolayer group. Bile reconstruction in the donor and recipient areas prevents bile congestion.

Retrorsine (RS) is a pyrrolizidine alkaloid that determines a persistent block of the endogenous hepatocyte cell cycle [19]. RS-induced hepatocellular injury severely impairs the capacity of fully differentiated rat hepatocytes to replicate. In humans, the intake of pyrrolizidine alkaloid-containing products is a major cause of hepatic sinusoidal obstruction syndrome (HSOS) [18]. In rat experiments, HSOS is associated with hepatic megalocytosis, ascites, hyperbilirubinemia, and hemorrhagic necrosis [20]. In a previous study, Yang et al. observed liver damage 24 h after RS treatment. In the high-dose RS (70–140 mg/kg) groups, the authors identified obvious liver damage with the following characteristics: dilation of sinusoids, destruction of central veins, and lobule disarray [21]. Of note, damage to sinusoidal endothelial cells, a characteristic of HSOS, was clearly demonstrated in the RS-treated lower dose (35 mg/kg). In the present study, we observed hepatocellular injury, ductular reaction, and liver fibrosis with lower-dose RS (15 mg/kg) combined with partial hepatectomy. 

In chronic and severe injury, ductular reactions of activated biliary epithelial cells containing hepatic progenitor cells (called oval cells) appear in the periportal area [21]. Hepatocytes, which are derived from oval cells, represent less than 4% of the total liver volume. Failure to remodel the extracellular matrix impairs the regenerative response. Ductular reactions and fibrosis are frequently associated. Additionally, under certain conditions, hepatic progenitor cells may contribute to profibrotic signals [22]. During ductular reaction, CK19-positive cells emerged. These cells differentiate to GST-p-positive precancerous cells [23]. Moreover, in the ductular reaction, there is the progressive formation of liver fibrosis [24]. In the present study, by transplanting liver organoids, ductular reaction is completely resolved. As a result, we observed a significant reduction of GST-p precancerous nodules compared with the sham operation group. 

The translocation of immature pluripotent stem cells outside the target organ is a dangerous situation, and may result in the development of tumors. Nicolas et al. studied the biodistribution of single-cell hepatocytes and hepatocyte spheroids in the liver, spleen, and digestive tract [17]. The pulmonary translocation of hepatocytes after transplantation has been described in several studies [25,26]. Muraca et al. showed that hepatocytes remained in the lung sinusoids for up to 48 h after infusion in a study involving pigs [25]. Bilir et al. reported that intraportal hepatocyte transplantation in patients with acute liver failure was followed by hypoxia and pulmonary infiltration, both of which improved after 24 h [27]. In the current study, the human iPS cell-derived organoids were well regulated, and no loose single cells were found surrounding the organoids. Additionally, no translocation of the liver organoid outside the liver was detected by PCR and immunohistochemistry. Alternatively, single cells constituting human iPS liver organoids were detected in the lungs after trans-portal injection. 

Vascularized and functional liver tissues generated entirely from human iPS cells significantly improved subsequent hepatic functionalization. This was enhanced by stage-matched developmental progenitor interactions, and decompensated liver function of acute liver failure mice was rescued by hiPSC liver organoids [9]. In the present study, we demonstrated that hiPSC liver organoids were attached to and repopulated the liver parenchyma and secreted human albumin.

## 4. Materials and Methods 

### 4.1. Animals and Retrorsine Administration

Three-week-old, female, DPPIV-negative Fischer 344/DuCrlCrlj rats were purchased from Charles River Laboratories Japan, Inc. (Yokohama, Japan). Female rats were selected because of their tolerance against surgical stress. DPPIV-positive Fischer 344/N Slc rats were purchased from Japan SLC (Shizuoka, Japan). The immunodeficient rats (F344-Il2rg^em7kyo^) were provided from Kyoto University (The National BioResource Project for the Rat in Japan No:0694). SD-Tg (CAG-EGFP) rats were provided from Japan SLC (Shizuoka, Japan). NOD SCID mice were provided from Sankyo Labo Service Corporation, Inc. (Tokyo, Japan). Two intraperitoneal RS injections (15 mg/kg body weight; Sigma Chemical Co., St. Louis, MO, United States) 2 weeks apart [19] were provided to the 4-week-old rats with DPPIV-negative livers. The dose was reduced because of its toxicity to the young rats. 

### 4.2. Isolation of Fetal Liver Cells and Liver Organoid Formation

Unfractionated fetal liver cells were isolated from ED14 fetal livers of pregnant DPPIV-positive F344 rats, as described previously [28]. For the liver organoid culture, harvested fetal liver cells were suspended in 10 mL William’s E medium (Thermo Fisher Scientific, Waltham, MA, United States), containing the following supplements: final concentration 10% Fetal Bovine Serum (MP Biomedicals, LLC, Irvine, CA, United States), 2 mM L-glutamine (Thermo Fisher Scientific), 1% penicillin/streptomycin (Thermo Fisher Scientific), 10 mM Nicotinamide (Sigma Chemical Co.), 50 μM 2-ME (β-ME) (Sigma Chemical Co.), 260 mM L-Ascorbic acid 2-phosphate sesquimagnesium salt hydrate (Sigma Chemical Co.), 5 mM HEPES (Dojindo Laboratories, Kumamoto, Japan), and 1 μg/mL human recombinant insulin expressed in yeast (FUJIFILM Wako Pure Chemical Corporation, Osaka, Japan). 

The cells were inoculated into 24-well Elplasia round-bottom-type plates (Kuraray, Tokyo, Japan) and cultured for 1 day, after which the diameter and number of self-organized rat liver organoids were analyzed (Keyence, Tokyo, Japan). Each liver organoid consisted of 5.0 × 10^3^ fetal liver cells. A monolayer fetal liver cell culture was performed on uncoated plastic dishes.

### 4.3. Liver Organoid and Monolayer Cell Transplantation to the Retrorsine/Partial Hepatectomy Liver

Four weeks after the second retrorsine (RS) injection, a two-thirds partial hepatectomy (RS/PH) was performed, as described previously [19]. Subsequently, monolayer cells (5.6 × 10^5^ cells, which is the equivalent of 3.0 × 10^3^ fetal liver organoid after a 1 day culture) or liver organoids (3.0 × 10^3^ organoids) from fetal liver cells were transplanted into the RS/PH livers (DPPIV-negative) through the main trunk of the portal vein using a 26G needle. The RS/PH procedure was performed without transplantation in the sham-operated group, whereas in the human iPSC liver organoid experiment, a 30% partial hepatectomy was performed with an RS injection. A two-thirds partial hepatectomy caused severe liver failure that was fatal to the immunodeficient rats. 

### 4.4. Liver Repopulation with the Transplanted Liver Organoids

The rats were sacrificed at different time points following the liver organoid transplantation. Subsequently, the liver tissues were embedded into the optimal cutting temperature compound and stored at −80 °C. The liver repopulation was determined by enzyme immunohistochemistry for DPPIV [28]; the repopulation rate was calculated by dividing the ratio of the implanted area by the liver area of the recipient. Measurements were performed using a Biozero 9000 microscope (Keyence).

### 4.5. Human Induced Pluripotent Stem Cell-Derived Liver Organoid Formation

The human iPS cell line (Ff-I01s04) was maintained on Laminin 511 E8-fragment-coated (iMatrix-511, kindly provided by Nippi) dishes in StemFit AK02N (Ajinomoto, Tokyo, Japan). Hepatic endoderm, endothelial, and mesenchymal cells were generated as described previously [29]. Next, human liver organoids were generated in vitro. To this end, a total of 900 cells per microwell, at a ratio of 10:7:1 (hiPSC-HE/iPSC-EC/iPSC-STM), were resuspended in a mixture of endothelial cell growth medium and hepatocyte culture medium (Cambrex, Baltimore, MD, United States). Additionally, dexamethasone (0.1 mM; Sigma-Aldrich, St. Louis, MO, United States), oncostatin M (10 ng/mL; R&D Systems, Minneapolis, MN, United States), and SingleQuots (Lonza Walkersville, Inc. Walkersville, MD, United States) were added to the medium. The cells were plated on a six-well Elplasia platform (co-developed by Kuraray, Tokyo, Japan), and the hiPSC liver organoids were cultivated from day 1 to day 12. The immunodeficient rat received daily injections (intraperitoneal) of 1 mg/kg of Prograf (Astellas Pharma Tech Co., Ltd. Toyama, Japan) every day, starting from one day before the transplantation of the hiPSC liver organoids. Furthermore, 10 mg/kg of Elaspol (Ono Pharmaceutical Co. Ltd. Osaka, Japan) and 20 μL heparin sodium (Mochida Pharmaceutical Co. Ltd. Tokyo, Japan) was added with the hiPSC liver organoids at the time of transplantation. Subsequently, 3.0 × 10^3^ hiPSC liver organoids were transplanted to the immunodeficient rats treated with RS/PH.

### 4.6. Histochemistry and Immunohistochemistry

The following primary antibodies were used in this study: CD26 mouse immunoglobulin (IgG) 2a (BD Pharmingen, San Jose, CA, United States); CD31 mouse IgG1 (BD Pharmingen); cytokeratin 19 mouse IgG2b (CK19; Progen, Heidelberg, Germany); GST-p rabbit IgG (MBL CO., LTD, Nagoya, Japan); HNF4a goat IgG (Santa Cruz Biotechnology, Dallas, TX, United States); hepatic sinusoidal endothelial cell mouse IgG2a (SE-1; IBL CO., LTD, Gunma, Japan); human vimentin mouse immunoglobulin (IgG) (DAKO: Agilent technologies, Santa Clara, CA, United States); human albumin (SIGMA) mouse IgG2a; human CD31 (DAKO) mouse IgG1; human CK19 (DAKO) mouse IgG1; and CK19 mouse IgG2b (Progen, Heidelberg, Germany). The second antibodies used were Alexa Fluor 488 goat anti-mouse IgG1(γ1) (Invitrogen: Thermo Fisher Scientific, Waltham, MA, United States), Alexa Fluor 647 goat anti-mouse IgG1 (γ1) (Invitrogen), Alexa Fluor 555 goat anti-mouse IgG2a (γ2a) (Invitrogen), Alexa Fluor 488 goat anti-mouse IgG2b (γ2b) (Invitrogen), Alexa Fluor 647 goat anti-rabbit IgG (H+L) (Invitrogen), and Alexa Fluor 488 goat anti-guinea pig IgG (H+L) (Life technologies: Thermo Fisher Scientific, Waltham, MA, United States). 

### 4.7. RT-PCRs

Total RNA was extracted from the rat fetal liver cell monolayers and the liver organoids using Isogen reagent (Nippon Gene Co., Ltd., Toyama, Japan), according to the manufacturer’s protocol. cDNA was synthesized from 1 mg of total RNA using the high-capacity cDNA Reverse Transcription Kit (Applied Biosystems, CA, United States). The cDNA was amplified with SYBR Premix ExTaq TMII (Takara Bio Inc. Shiga, Japan), using the ABI PRISM 7700 (Applied Biosystems, Foster City, CA, United States) system. Quantitative polymerase chain reaction (PCR) was performed with ABI TaqMan Gene Expression Assays (Applied Biosystems, Foster City, CA, United States) for rat albumin, HNF4a, CK19, and g6p; 18s rRNA was used as the control. The primers used in this study were as follows: *18S rRNA* forward (FW) = AAGTTTCAGCACATCCTG CGAGTA; reverse (RW) = TTGG TGAGGTCAATGTCTGCCTTTC; *Alb* FW = CACCAAATTGGCAACAGACGT TAC; RW = CAAGCCTGCAGTTTGCTGGA; *Krt19* FW = TCAGTATGAGGCCA TGGCAGAG; RW = CCTTACGTCGGAGTTCCGTGA; *HNF4a* FW = GGCTGG CA TGAAGAAAGAAG; RW = GAGCGCATTAATGGAGGGTA; and *g6p* FW = AAC GTCT GTCTGTCCCGATCTAC; RW = ACTCTGGAGGCTGGCATTG.

The genomic DNA of dissected samples of the liver, lung, kidney, spleen, brain, and heart from NOD SCID mice were purified. The genomic DNA was amplified with the following primer sequences: *Human Alu* FW = CGAGGCGG GTGGATCATGAGGT; RW = TCTGTCGCCCA GGCCGGACT; mouse *GAPDH* FW = ATCATCTCCGCCCCTTCTGC; RW = TGAGCCCTTCCACAATGCCA. 

### 4.8. Serological Analyses

Whole blood samples were centrifuged (4000 revolutions per min) for 20 min at 4 °C, and the supernatants were collected and stored at −30 °C. Subsequently, the supernatants were tested using a serum multiple biochemical analyzer (Fuji Drichem; Fuji Film Inc., Tokyo, Japan). The human albumin level in the rat blood was assessed by enzyme linked immunosorbent assay (Bethyl Laboratories, Montgomery, TX, United States), according to the manufacturer’s instructions.

### 4.9. Study Approval

We bred and maintained the rats according to our institutional guidelines for the care and use of laboratory animals. All animal studies were carried out following approval from the Institutional Animal Care Use Committee of Yokohama City University (approval nos. 17–25 from 1st April, 2017).

### 4.10. Statistics

All data are expressed as mean ± standard deviation (SD). Comparisons between the two groups were performed using the Mann–Whitney *U* test, whereas comparisons between various points were made using one-way analysis of variance (ANOVA). Significant data were examined using the Bonferroni–Dunn multiple comparisons post-hoc test. A *p*-value < 0.05 was considered significant.

## 5. Conclusions

In conclusion, this study clearly demonstrates that liver organoids can be safely transplanted through the portal vein. The fetal liver organoids were superior to monolayer cells in terms of liver regeneration and bile duct reconstruction. Additionally, these liver organoids can completely substitute chronically damaged liver. Furthermore, they can drastically decrease precancerous lesions. The hiPSC liver organoids can be safely transplanted through the portal vein without translocating to the other organs. This study precedes a theoretical treatment model to substitute organ transplantation with liver organoid transplantation.

## Figures and Tables

**Figure 1 ijms-21-00178-f001:**
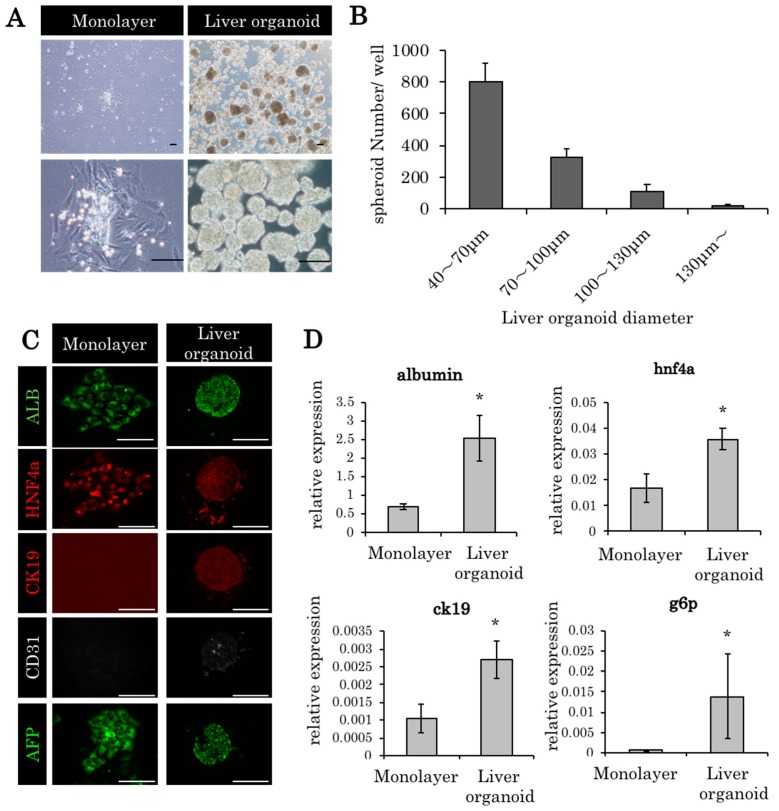
Liver organoids made from embryonic day 14 (ED14) rat fetal liver cells. (**A**) Monolayer culture and liver organoids derived from embryonic day (ED) 14 fetal liver cells (culture period, 24 h). Scale bars: 100 μm. (**B**) The diameters of most of the liver organoids were between 40 and 70 μm. (**C**) The histopathological findings of the monolayer culture and liver organoids derived from ED14 fetal liver cells (ALB: albumin; HNF4a: hepatocyte nuclear factor 4 alpha; and CK19: cytokeratin 19). Scale bars: 100 μm. (**D**) Gene expression in monolayer culture and liver organoids 24 h after incubation. In glucose 6 phosphatase (g6p), 96 h culture specimens were used. The relative expression levels of the genes compared to 18s ribosomal RNA are indicated by PCR. * *p* < 0.05 vs monolayer, Mann–Whitney *U* test, *n* = 3.

**Figure 2 ijms-21-00178-f002:**
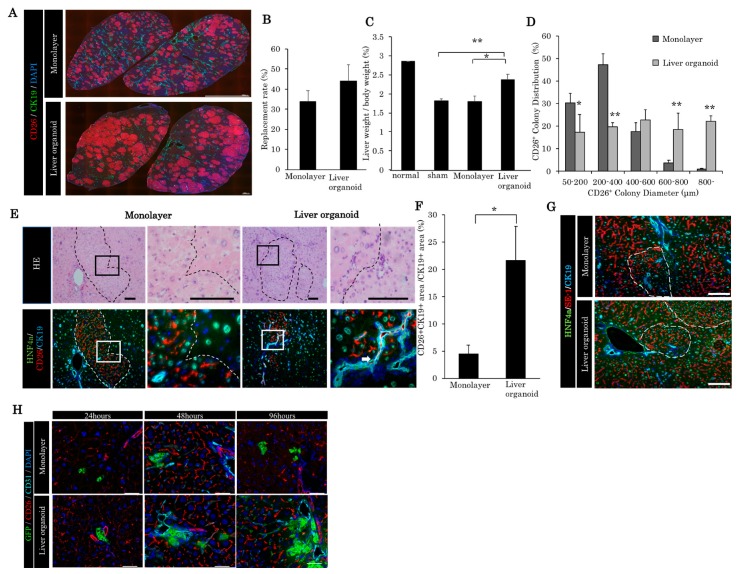
Comparison of liver regeneration and bile duct reconstruction between monolayer cells and liver organoids in the chronic liver damaged by retrorsine/partial hepatectomy (RS/PH). (**A**) Repopulation of transplanted cells in the retrorsine/partial hepatectomy (RS/PH) models. A total of 3.0 × 10^3^ liver organoid monolayer cells, the equivalent of 3.0 × 10^3^ liver organoids, were transplanted. Thirty days after transplantation, quadrate lobes were immunohistochemically stained with CD26, CK19, and 4′,6-diamidino-2-phenylindole (DAPI). (**B**) Repopulation rate of the transplanted cell in the quadrate lobe of the RS/PH liver 30 days after transplantation. (**C**) Liver weight/body weight ratios of the normal liver and of those in the sham operation group, monolayer cell group, and liver organoid group 30 days after transplantation. * *p* < 0.05, ** *p* < 0.01 vs liver organoid group; one-way ANOVA with Bonferroni correction. (**D**) Distribution of CD26+ colony diameter. * *p* < 0.05, ** *p* < 0.01 vs monolayer group; Mann–Whitney *U* test, *n* = 3. (**E**) The donor-derived bile duct was anastomosed to the recipient bile duct. White arrows indicate the anastomosis. In the monolayer cell transplanted group, the donor-derived bile ducts could not be observed. Scale bars: 100 μm. (**F**) The ratio of the CD26+/CK19+ area to the CK19+ area showing the donor-derived bile duct area. * *p* < 0.05 vs monolayer, Mann–Whitney *U* test, *n* = 3. (**G**) The liver sinusoidal endothelial cells were observed both in the donor area and recipient area. SE-1: liver sinusoidal endothelial cell marker. Scale bars: 100 μm. (**H**) Twenty-four, 48, and 96 h after transplantation of enhanced green fluorescent protein (EGFP)-positive fetal rat-derived liver organoids and monolayer cultured cells. Note that liver organoids can spread faster than monolayer cells from the periphery of the portal branches without portal vein thrombus. Scale: 100 μm.

**Figure 3 ijms-21-00178-f003:**
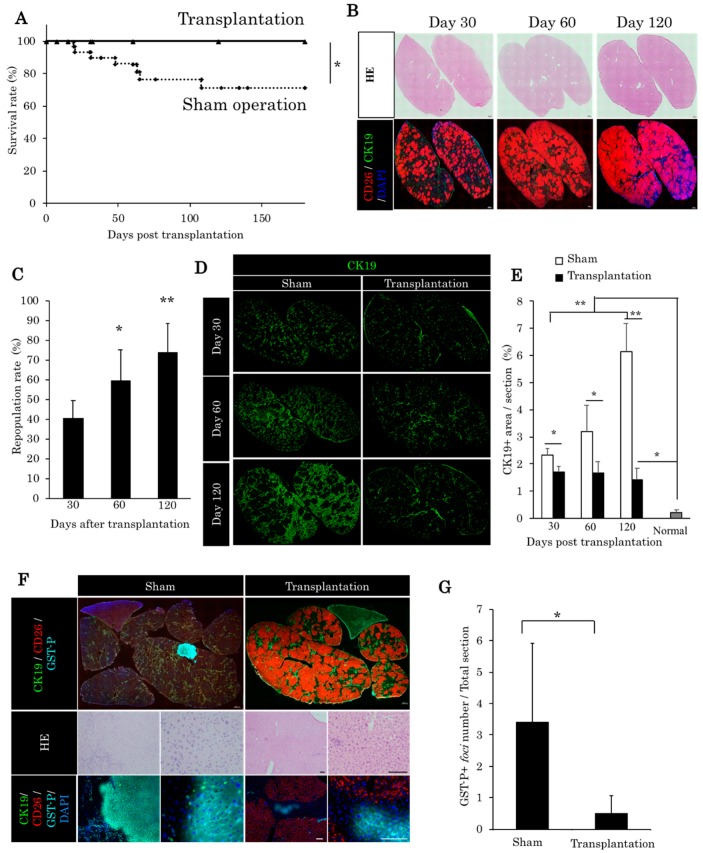
Therapeutic effect of liver organoids in chronic liver damage from RS/PH. (**A**) Survival rate of the sham and transplantation group post-transplantation. * *p* = 0.024332 (log rank test). Sham group *n* = 22. Transplantation group *n* = 17. (**B**) DPPIV/CD26-positive liver organoids repopulated in the RS/PH liver of the DPPIV-negative rats. A total of 3.0 × 10^3^ liver organoids was injected from the portal vein. Images of the quadrate lobe. Thirty, 60, and 120 days post-transplantation are shown. Scale: 1 mm. (**C**) Repopulation rate of the recipient liver by DPPIV/CD26-positive liver organoids. * *p* < 0.05, ** *p* < 0.01 vs. day 30 (one-way ANOVA with Bonferroni correction). (**D**) CK19 expression of the quadrate lobe. In the sham group, the area of CK19 increased between 30 and 120 days after RS/PH treatment. On the contrary, the CK19-positive area in the transplantation group did not increase. (**E**) CK19 area in the quadrate lobe. Thirty to 120 days post-transplantation, CK19 areas of the transplantation group were significantly decreased compared with the sham group. Bonferroni correction and Mann–Whitney *U* test * *p* < 0.05, ** *p* < 0.01 vs. sham group (*n* = 3–4). (**F**) placental glutathione S-transferase (GST-p) foci of the two groups. Large GST-p-positive foci were observed in the sham group 120 days after RS/PH treatment. (**G**) Numbers of GST-p-positive foci in the total section of the liver. In the transplantation group, the number of GST-p-positive foci was significantly lower compared with the sham group. * *p* < 0.05 vs. sham group (Mann–Whitney *U* test, *n* = 3–4).

**Figure 4 ijms-21-00178-f004:**
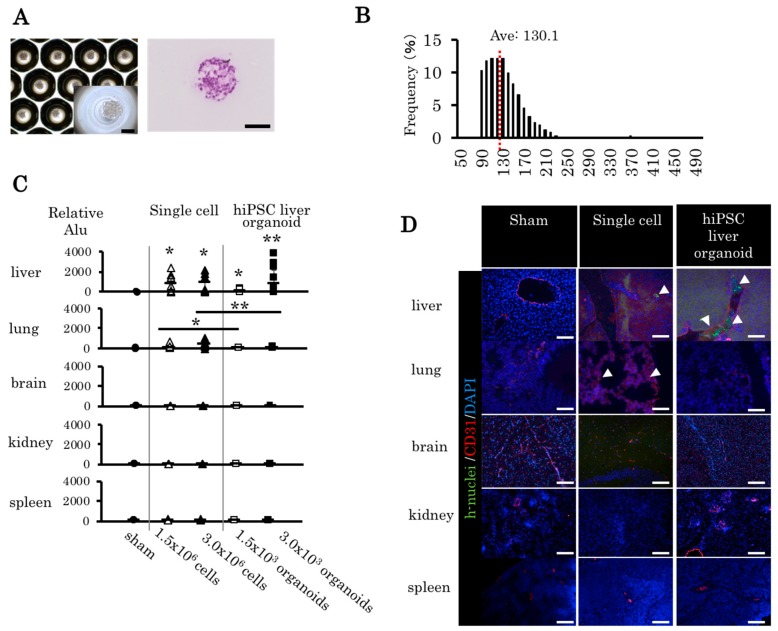
Bio-distribution of human induced pluripotent stem cell (hiPSC) liver organoids and single-cell transplantation through the portal vein. (**A**) HiPSC liver organoid in the micropattern plate with bright field and hematoxylin and eosin (HE) staining. (**B**) Distribution of diameter (average = 130.1 μm). (**C**) Human cell DNA in each organ was detected in the cell transplantation group and the liver organoid transplantation group. Mean ± SD, *n* = 6–14, one-way ANOVA and Bonferroni–Dunn test; * *p* < 0.05, ** *p* < 0.01. (**D**) The liver, lung, brain, kidney, and spleen from each transplant group were removed and histologically analyzed. Green: human nuclei, red: murine CD31, blue: DAPI. Arrowhead: human cells, magnification 20×. Scale bars: 100 μm.

**Figure 5 ijms-21-00178-f005:**
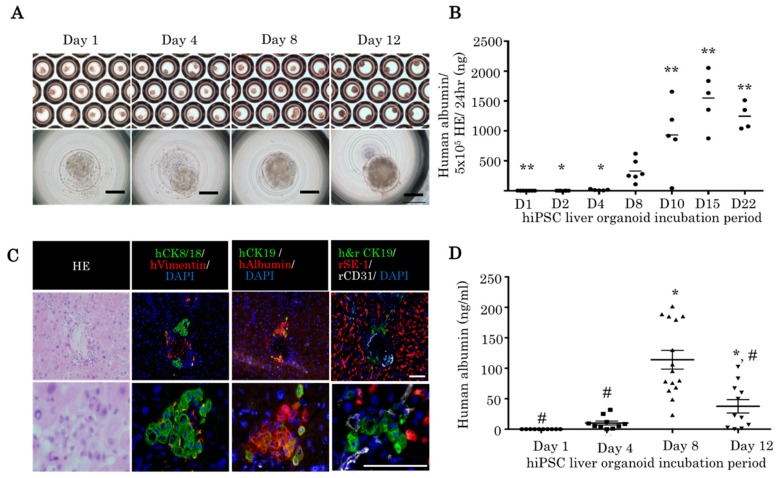
HiPSC liver organoids attached to the immunodeficient rat liver via portal vein injection. (**A**) HiPSC liver organoid in the micropattern plate after 1, 4, 8, and 12 days of incubation (bright field). Scale bars: 100 μm. (**B**) Human albumin was detected in the supernatant of hiPSC liver organoids (one-way ANOVA with Bonferroni correction, ** *p* < 0.01, * *p* < 0.05 vs. day 8 hiPSC liver organoid). (**C**) Histopathological examination of hiPSC liver organoid transplanted rat liver 14 days post-transplantation. Reconstituted human liver tissue expressing human vimentin, human CK8/18, human albumin, human CD31, human CK19, and human and rat CK19. Scale bars: 100 μm. (**D**) Human albumin was detected in the rat serum 14 days post-transplantation. A total of 3 × 10^3^ liver organoids after 1, 4, 8, and 12 days of incubation were transplanted (one-way ANOVA with Bonferroni correction, * *p* < 0.01 vs. day 1; # *p* < 0.01 vs. day 8).

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
