# Peer review of "The Regenerative Effect of Portal Vein Injection of Liver Organoids by Retrorsine/Partial Hepatectomy in Rats"

_ijms, 2019, doi:10.3390/ijms21010178_

Round 1

Reviewer 1 Report

In the current manuscript, Tsuchida et al. show the feasibility and efficiency of liver organoid transplantation from fetal rat liver and human iPSC in a model of liver resection on injured livers.

The authors should be congratulated for this work for several reasons:

-first, the model is clinically relevant as hepatectomy is realized on injured livers which is the actual clinical setting but which is often not the case in experimental studies.

-second, it shows a valuable benefit after hepatectomy, focusing on tissue repair (bile duct formation notably) and not just liver regeneration which seems more clinically relevant, once again. 

The main question raised by the manuscript are regarding this second point:

- in regard to liver tissue repair, the preservation of functional sinusoidal endothelial cells with persistent fenestration has been shown of utmost importance and no data is shown regarding this point.

- the higher liver weight/body weight ratio may not be necessarily a sign of more important regeneration but may be explained by hypertrophy rather than hyperplasia which is in line with the non significant increase in liver repopulation ratio in the organoid group vs monolayer cells group. Have the authors looked at cell ploidy?

- the number of experiments (n=3-4) is quite small in view of the potential impact of the data shown herein

Minor questions

- May the authors precise why they used both models with fetal rat liver and human iPSC?

- The authors state that there is fear for portal vein thrombosis after organoid injection. Is this published data or theoretical hypothesis ? It may be infered that there is a specificity in their liver organoid preparation that explains the tolerance of organoids in the portal vein in their experiments. May they comment on that point?

- Their is a slight inconsistency in the method section which states that "each liver organoid consisted of 5000 fetal liver cells" and then that 3x10^3 cells are transplanted per liver organoid.

- Introduction should be rewritten in a shorter and more logical way. 

Altogether, this manuscript is of high quality and may be a breakthrough in cellular bioengineering in the field of hepatology and liver surgery. 

Author Response

Reviewer 1

In the current manuscript, Tsuchida et al. show the feasibility and efficiency of liver organoid transplantation from fetal rat liver and human iPSC in a model of liver resection on injured livers. The authors should be congratulated for this work for several reasons:-first, the model is clinically relevant as hepatectomy is realized on injured livers which is the actual clinical setting but which is often not the case in experimental studies.-second, it shows a valuable benefit after hepatectomy, focusing on tissue repair (bile duct formation notably) and not just liver regeneration which seems more clinically relevant, once again. 

The main question raised by the manuscript are regarding this second point:

- in regard to liver tissue repair, the preservation of functional sinusoidal endothelial cells with persistent fenestration has been shown of utmost importance and no data is shown regarding this point. 

→We would like to thank reviewer 1 for addressing this important comment. According to the reviewer 1`s comment, we added the preservation of endothelial cell lining both in donor and recipient area (Fig 2G).

- the higher liver weight/body weight ratio may not be necessarily a sign of more important regeneration but may be explained by hypertrophy rather than hyperplasia which is in line with the non significant increase in liver repopulation ratio in the organoid group vs monolayer cells group.Have the authors looked at cell ploidy?

→Thank you for the important comment. We understood the importance of cell ploidy and we will try to analyze the cell ploidy later.

- the number of experiments (n=3-4) is quite small in view of the potential impact of the data shown herein

→Thank you for the comments. In Figure 5, we performed more than 10 animals. In Figure 4, we tested more than 6 animals. In Figure 3, we checked survival rate in 39 animals. But In figure 1~3 we apology the number is small. We should try to increase the number later.

Minor questions

- May the authors precise why they used both models with fetal rat liver and human iPSC?

→Thank you for the important comment. We think the fetal liver organoid experiment is for the revealing the proof of the concept, whereas human iPSC liver organoid experiment stands for the preclinical experiment.

-The authors state that there is fear for portal vein thrombosis after organoid injection. Is this published data or theoretical hypothesis ? It may be inferred that there is a specificity in their liver organoid preparation that explains the tolerance of organoids in the portal vein in their experiments. May they comment on that point?

→Thank you for the very important comments. In the revised manuscript, we added the explanation in line 54 to 59 and references (11, 12, 13) of portal vein thrombus after hepatocyte transplantation from the portal vein. The size of hepatocyte is around 20um, whereas liver organoid is more than 130um and the risk of portal vein thrombus should be increased.

- There is a slight inconsistency in the method section which states that "each liver organoid consisted of 5000 fetal liver cells" and then that 3x10^3 cells are transplanted per liver organoid.

→Thank you for the comment. In line 286, we described each liver organoid consist of 5,000 fetal liver cells. In line 290, we explained the transplantation 3x103liver organoids.

-Introduction should be rewritten in a shorter and more logical way. 

→Thank you for the important comment. According to the reviewer 1`s mindful comment, we edited the introduction more shorter.

Altogether, this manuscript is of high quality and may be a breakthrough in cellular bioengineering in the field of hepatology and liver surgery. 

Reviewer 2 Report

The authors investigated in this work the effects of liver organoid transplantation through the portal vein in regards to safeness and effectiveness as treatment of chronic liver damage in rats. The manuscript reports potentially interesting data, it is clearly written and well explained.

Specific comments: 

I strongly recommend adjusting the conclusion of the abstract to include the model organism: “This study clearly demonstrated that liver organoid transplantation through the portal vein is a safe and effective method for the treatment of chronic liver damagein rats. 
 Please correct the spelling and phrasing, i.e. lines 72-73: Albumin, HNF4alpha, and CD31 was expressed at higher levels in the liver organoid culture express compared to the monolayer culture; line 150: “mean diameter of the organoid was 130.1 m (Figure 4B)” – 130 meters? 
 Line 88: how many cells / liver organoids were transplanted? Would the difference in weight be justified by the weight of the grafted organoids? (figure 2C). Are the single human iPS cell derived cells relocated in different organs forming tumors at later timepoints? 
 How would the authors explain the decrease of human albumin in rat serum at day 12? Are the iPSC-derived grafts stable in time?

Author Response

Reviewer 2

The authors investigated in this work the effects of liver organoid transplantation through the portal vein in regards to safeness and effectiveness as treatment of chronic liver damage in rats. The manuscript reports potentially interesting data, it is clearly written and well explained.

Specific comments: 

I strongly recommend adjusting the conclusion of the abstract to include the model organism: “This study clearly demonstrated that liver organoid transplantation through the portal vein is a safe and effective method for the treatment of chronic liver damage” in rats.

→Thank you for the very important comments. According to the reviewer 2`s comment, we corrected the conclusion as This study clearly demonstrated that liver organoid transplantation through the portal vein is a safe and effective method for the treatment of chronic liver damage in rats”.

Please correct the spelling and phrasing, i.e.

lines 72-73: Albumin, HNF4alpha, and CD31 was expressed at higher levels in the liver organoid culture express compared to the monolayer culture; corrected.

 line 150: “mean diameter of the organoid was 130.1 m (Figure 4B)” – 130 meters? 
Thank you for the comment. According to the reviewer’s comment, we corrected the unit in line 162.

  Line 88: how many cells / liver organoids were transplanted?

→Thank you for the comment. According to the reviewer’s comment, we added the number in line 93-94.

Would the difference in weight be justified by the weight of the grafted organoids?

(figure 2C).

→Thank you for the comment. The weight of the cells is too small and we did not correct the weight of the grafted organoid.

Are the single human iPS cell derived cells relocated in different organs forming tumors at later timepoints? 

→Thank you for the important comment. At the later timepoints, we did not detect relocation of cells in different organs.

How would the authors explain the decrease of human albumin in rat serum at day 12? Are the iPSC-derived grafts stable in time? 

→Thank you for the important comment. We added the appearance of the day 12 hiPSC liver organoid and the albumin secretion of the organoids in vitro in Figure 5A. The hiPSC derived liver organoid is stable with more than 12 day culture. The reason of the decrease of human albumin of the 12 day cultured hiPSC liver organoid in vivo is the decreased ability of attachment and proliferation in vivo after transplantation.

Round 2

Reviewer 1 Report

The authors have answered to most of the questions.

However one important is still pending and should be precisely addressed in the final manuscript : could the authors explain what they believe is the reason for not observing portal vein occlusion in their specific model ? Is there any pre- or post-treatment ?

Minor comments : the 2 sentences at the end of the introduction (line 63-66) concerning the model (DPPIV and retrorsine) do not fit at this place and should be deleted in my point of view.

Author Response

Reviewer 1

The authors have answered to most of the questions.

However one important is still pending and should be precisely addressed in the final manuscript : could the authors explain what they believe is the reason for not observing portal vein occlusion in their specific model ? Is there any pre- or post-treatment ?

→We would like to thank reviewer 1 for addressing this important comment. According to the reviewer 1`s comment, we added the distribution of the transplanted organoids in the early 96 h post-transplantation, and the figure shows the liver organoids disassembled and spread faster than monolayer culture around the sinusoidal area without portal thrombus (Figure 2H). We think the fragility and the ability to spread of the liver organoids avoid portal vein occlusion after transplantation.  

Minor comments : the 2 sentences at the end of the introduction (line 63-66) concerning the model (DPPIV and retrorsine) do not fit at this place and should be deleted in my point of view.

→Thank you for the important comment. In the revised manuscript we deleted the 2 sentences according to the reviewer 1`s comment.